# Diverse Mechanisms of Antimicrobial Activities of Lactoferrins, Lactoferricins, and Other Lactoferrin-Derived Peptides

**DOI:** 10.3390/ijms222011264

**Published:** 2021-10-19

**Authors:** Špela Gruden, Nataša Poklar Ulrih

**Affiliations:** Department of Food Science and Technology, Biotechnical Faculty, University of Ljubljana, 1000 Ljubljana, Slovenia; Spela.Gruden@bf.uni-lj.si

**Keywords:** lactoferrin, lactoferricin, peptides, antimicrobial activity, mechanisms of action

## Abstract

Lactoferrins are an iron-binding glycoprotein that have important protective roles in the mammalian body through their numerous functions, which include antimicrobial, antitumor, anti-inflammatory, immunomodulatory, and antioxidant activities. Among these, their antimicrobial activity has been the most studied, although the mechanism behind antimicrobial activities remains to be elucidated. Thirty years ago, the first lactoferrin-derived peptide was isolated and showed higher antimicrobial activity than the native lactoferrin lactoferricin. Since then, numerous studies have investigated the antimicrobial potencies of lactoferrins, lactoferricins, and other lactoferrin-derived peptides to better understand their antimicrobial activities at the molecular level. This review defines the current antibacterial, antiviral, antifungal, and antiparasitic activities of lactoferrins, lactoferricins, and lactoferrin-derived peptides. The primary focus is on their different mechanisms of activity against bacteria, viruses, fungi, and parasites. The role of their structure, amino-acid composition, conformation, charge, hydrophobicity, and other factors that affect their mechanisms of antimicrobial activity are also reviewed.

## 1. Introduction

Lactoferrins are iron-binding proteins that belong to the transferrin family. Since the first isolation of lactoferrins from both bovine [1] and human [2,3] milk in 1960, they have been the subject of intensive structural and functional studies, especially because of their numerous functions, properties, and applications in the food and pharmaceutical industries. Lactoferrins have also been identified in other mammalian species, as listed in Table 1; however, bovine and human lactoferrins have been the most studied to date.

Lactoferrins are produced by the epithelial cells of different body organs [4], and as a consequence, they are found in numerous exocrine secretions, including the colostrum, milk, tears, nasal and bronchial secretions, saliva, bile and pancreatic secretions (and therefore in gastric and intestinal fluids), urine, and seminal and vaginal fluids [5]. Furthermore, lactoferrins are also produced by the hematopoietic tissue of bone marrow and are found in granules of polymorphonuclear neutrophils [4,6]. Colostrum and milk are the most abundant sources of lactoferrins, where the concentrations depend on the mammalian species and the stage of lactation [4]. Lower concentrations of lactoferrins are found in other exocrine secretions. Additionally, for neutrophils, the lactoferrin concentrations vary across mammalian species [7], while in humans, the structure of the neutrophil lactoferrin is identical to that from milk [8].

Lactoferrins [14,15,16] are monomeric glycoproteins with a molecular mass of ~80 kDa, and their three-dimensional structures and amino-acid sequences are now well established. The primary amino-acid sequences of lactoferrins from different mammalian species have been reported, including those for human [15,17], cow [18,19], goat [20,21], sheep [13], camel [22], buffalo [23], horse [24], pig [25], and mouse [26]. Bovine lactoferrin consist of a single polypeptide chain of 689 amino acids, while human lactoferrin has two more, at 691 amino acids.

Anderson et al. (1987, 1989) reported the first crystallographic structure of human lactoferrin [27,28]. To date, the crystallographic structures of lactoferrins have been determined for four other species: cows [17], camels [22], buffalos [23,29], and horses [24], with small differences seen for the structures between species. The single polypeptide chain of lactoferrins is folded into two globular lobes that represent the N-terminal and C-terminal halves of the polypeptide, which share ~40% sequence homology [27]. These two lobes are thus referred to as the N-lobe (amino acids 1–333) and C-lobe (amino acids 345–689/691), and they are further divided into two domains each, as N-1 (amino acids 1–90, 251–333), N-2 (amino acids 91–250), C-1 (amino acids 345–431, 593–689), and C-2 (amino acids 432–592). The N-lobe and C-lobe are connected by a short, three-turn α-helix as amino-acids 334 to 344. Each pair of N-1 and N-2, and C-1 and C-2 domains encloses a deep cleft that provides the iron binding site. This iron binding site is comprised of four amino acids (Tyr [×2], Asp, His) that provide three negative charges to balance the 3+ charge of Fe^3+^, together with a helix at their N-terminus and an arginine sidechain, the positive charge of which balances the negative charge on a CO_3_^2-^ anion. Based on iron saturation, lactoferrins can adopt two conformations, as the iron-saturated and more closed holo-lactoferrins, and the iron-free and more open apo-lactoferrins (Figure 1). Each lactoferrin molecule can bind two iron ions, both reversibly and strongly (K ≈ 10^22^ M), together with two synergistically bound CO_3_^2−^ ions [17,30,31,32,33]. Involvement of CO_3_^2-^ anion at the iron binding site appears to have a role in iron release at acid pH [30,34]. Iron binding gives lactoferrins a specific red color and was also referred to as red milk protein in the past [1,34].

All proteins go through post translation modification (PTM) during their maturation, and lactoferrin is no exception. Glycosylation is one of the most common PTMs, and glycans are crucial macromolecules for all living organisms [35]. Lactoferrins from all species are glycosylated; however, the number of potent glycosylation sites vary between species. Human lactoferrin have three potential N-glycosylation sites (asparagine (Asn) 138, Asn479, and Asn624) while bovine lactoferrin have five (Asn233, 281, 368, 476, and 545), but only two sites are usually glycosylated in human lactoferrin (Asn137 and Asn478) and four are glycosylated in bovine lactoferrin (Asn233, Asn368, Asn476, and Asn545) [31,36,37].

Numerous functions have been attributed to lactoferrins, which have ranged from antimicrobial (i.e., antibacterial, antivirus, antifungal, and antiparasitic) to antitumor, anti-inflammatory, immunomodulatory, and antioxidant activities.

Antibacterial activity was the first of these to be ascribed to lactoferrins, which have been well studied [38,39,40], since they represent a great potential as a natural defense agent. Lactoferrin and lactoferrin-derived peptides not only have a broad specter of antibacterial activities against Gram-positive and Gram-negative bacteria and can be potentially used as natural antibiotic in human and veterinary medicine [38] but also have a broad spectrum of activities against enveloped and naked viruses [41,42,43], fungi, yeast [44,45], and parasites [45]. Furthermore, lactoferrin has also been proposed to be potent as a treatment drug in the current COVID-19 pandemic caused by severe acute respiratory syndrome coronavirus 2 (SARS-CoV 2) [46,47,48,49,50]. The antimicrobial activity of lactoferrin is related to its ability to chelate iron and thus to deprive microorganisms of these important nutrient, although lactoferrin also expresses antimicrobial activity in the iron-independent pathway by direct interaction with microorganisms.

The lactoferrin iron-binding ability also impacts immune homeostasis and lactoferrin anti-inflammatory function. Since iron is crucial in modulating the production of reactive oxygen species (ROS), lactoferrin as an iron-binding protein can reduce oxidative stress caused by reactive oxygen species and can thus control excess inflammatory response. Furthermore, lactoferrin has a strong modulatory effect on the innate and adaptive immune responses by accelerating the maturation of T-cell and by differentiating immature B-cells. Moreover, during inflammation, lactoferrin exerts anti-inflammatory activity against IL-6 [51,52,53,54].

Despite recent advances in cancer therapy, current treatments present attendant side effect and also affect the quality of life of patients. Lactoferrin and lactoferrin-derived peptides also show a great potential in cancer treatment because of its diverse anticancer properties such as arresting cancer cell growth; blocking cancer cell cycle; inducting cell apoptosis; and inhibiting cancer cell migration, invasion, and metastasis. Furthermore, lactoferrin and lactoferrin-derived peptides shows anticancer activity against different types of cancer and can be easily orally administrated or potentially used for cancer drug transport [55,56].

As lactoferrins were shown to be in body fluids that usually interact with the surrounding environment and considering their broad activity against different microbes, it is initially believed that lactoferrins have an important role in the initiation of the immune system.

The enzymatic digestion of bovine lactoferrin with pepsin by Tomita et al. in 1991 led to the discovery of a cleavage peptide, further named bovine lactoferricin (17–41) [57], which showed improved antibacterial activity over the native lactoferrins [58]. Since then, numerous peptides with antimicrobial activities have been produced by enzymatic digestion of lactoferrins or have been synthesized. Studying these peptides might enable us to understand the mechanisms of this antimicrobial activity at the molecular level.

This review discusses the antibacterial, antiviral, antifungal, and antiparasitic activities of lactoferrins, lactoferricins, and other lactoferrin-derived peptides (i.e., shorter lactoferricin peptides). The focus is on the different mechanisms behind these activities, and the factors that affect their antimicrobial activities, which include their structure, amino-acid composition, conformation, charge, and hydrophobicity. The term lactoferrin-derived peptides is used to refer to peptides that are based on the lactoferrin molecule, with lactoferricin-derived peptides similarly referring to peptides that are based on the lactoferricin sequence, which are thus shorter versions of the lactoferricin peptides.

## 2. Antibacterial Activities of Lactoferrins and Lactoferrin-Derived Peptides

Antibacterial activity was one of the first functions ascribed to lactoferrins, and this has also been the most studied of their activities (Appendix A). Lactoferrins have two mechanisms of action against a great variety of Gram-positive and Gram-negative bacteria: bacteriostatic and bactericidal.

The bacteriostatic activity of lactoferrins arises through their ability to bind to iron, to thus deprive bacteria of this important nutrient, as bacterial growth can be restored by the simple addition of exogenous iron in excess of the chelating capacity of the lactoferrin [59]. The bactericidal mechanism of lactoferrins was first reported by Arnold et al. (1977) [60]. They demonstrated that, in an iron-rich medium, human lactoferrin inhibited the growth of *Vibrio cholerae* and *Streptococcus mutans* but not *Escherichia coli* and that this antibacterial activity was not reversed in the presence of supplemental iron. Based on immunofluorescence studies, they suggested that human lactoferrin binds to the surface of bacterial cells [60]. Further studies have shown that bovine lactoferrin [61] and human lactoferrin [62,63,64,65] bind and release lipopolysaccharides (LPS) from enteric Gram-negative bacilli and demonstrated that lactoferrins can interact directly with the bacteria cell membranes. Furthermore, bovine and human lactoferrin have the same binding site for LPS [65].

The outer membrane of Gram-negative bacteria is an asymmetric lipid bilayer with a phospholipid-rich inner leaflet and an outer leaflet that is predominantly comprised of LPS. As shown schematically in Figure 2, these LPS are typically composed of lipid A, a short core oligosaccharide, and an O antigen [66].

It has been shown that human lactoferrin can bind to lipid A of LPS, and for this binding, the Arg2–Arg3–Arg4–Arg5 stretch of human lactoferrin is essential [68,69]. The human lactoferrin binding site for LPS is in the N-terminal domain loop region of amino acids 20 to 37, and more specifically, this involves amino acids 28 to 34 (Figure 3A). This loop region is also present in the sequence of the lactoferrin cleavage peptide known as lactoferricin (amino acids 1–49; Figure 1). In bovine lactoferrin, this loop region corresponds to amino acids 19 to 36, which forms most of the bovine lactoferricin sequence (see below).

The bacteriostatic and bactericidal activities of lactoferrins have only been reported for the iron-free apo-lactoferrins (iron saturation in about 10–20%), which are the naturally occurring forms of lactoferrins in body fluids; indeed, the iron saturated form of holo-lactoferrins does not inhibit bacterial growth [60,62,71,72,73,74,75]. Furthermore, the release of LPS from Gram-negative bacteria outer membrane by lactoferrins can be blocked by high levels of divalent ions, such as Ca^2+^ and Mg^2+^ [62,63,64,73]. There are several explanations as to why the antibacterial activity of lactoferrins is lost in the presence of iron and divalent cations. One relates to the conformational changes that accompany the iron binding to lactoferrins. These conformation changes occur especially in the N-lobe, with the C-lobe hardly changed under iron saturation [30,33]. The in silico analysis of the interdomain movements by Lizzi et al. (2016) also showed that the conformation of the active N-terminal part of bovine apo-lactoferrin is more open than that of bovine holo-lactoferrin, with increased accessibility of the N-terminal to the bacterial membrane in the apo-conformation [75]. Of note, the N-terminal part of apo-lactoferrins includes the lactoferricin cleavage peptides and other known antibacterial peptides (see below), and this region includes the antibacterial domain of lactoferrins (Figure 1).

As indicated, changes in the N-terminal of lactoferrins upon Fe^3+^ binding are most likely to be crucial for this loss of antibacterial activity in the holo-conformation. Bennett et al. (1981) showed that conformational changes also occur in the presence of Ca^2+^, where the lactoferrin molecule can undergo polymerization to form tetramers [76]. The negatively charged saccharide headgroups of LPS molecules (i.e., the ‘inner core’ of oligosaccharides) are electrostatically linked by divalent cations, in particular by Ca^2+^ and Mg^2+^, which stabilizes the structural integrity of the bacteria and protects the bacteria from their environment [77,78,79]. It has been shown that metal chelators such as EDTA bind membrane-stabilizing cations such as Ca^2+^ and Mg^2+^, which results in the release of LPS and the disruption of the outer bacterial membrane. High concentrations of Ca^2+^ and Mg^2+^ block this release of LPS by EDTA [62,80], so a similar mechanism was proposed for lactoferrins. The binding of Ca^2+^ by bovine lactoferrin has been reported only by Rossi et al. (2002) [81]. The inhibition of lactoferrin antibacterial activities in the presence of high concentrations of Ca^2+^ and Mg^2+^ might also occur because of stabilization of the outer bacterial membrane by these cations to prevent the release of LPS [79]. There have been various studies on the effects of Mg^2+^ ions on the antibacterial activity of lactoferrin. Arnold et al. (1981) reported that antibacterial activity of human lactoferrin was not affected by Mg^2+^ ions [72], while Kalmar et al. (1988) showed that >1 mM Mg^2+^ enhanced human lactoferrin killing of *Aggregatibacter actinomycetemcomitans* (previously *Actinobacillus actinomycetemcomitans*) [72,74]. As well as these effects of Ca^2+^ and Mg^2+^, the lactoferrin antibacterial activities are also affected by pH, temperature, and buffer. Lactoferrins shows their highest activities at slightly acid pH (5.0–6.0) and temperature (37 °C) [72,74].

Another aspect of the lactoferrin antimicrobial activity might arise from the binding of bovine lactoferrin to porins. Porins are transmembrane proteins that form channels for nonspecific diffusion of hydrophilic solutes across the outer membrane of Gram-negative bacteria [66]. Binding of bovine lactoferrin to porins OmpF and OmpC has been demonstrated [82,83].

Lactoferrins also have other mechanisms of antibacterial activity that might be important. For example, bovine lactoferrin downregulates the *luxS* gene that encodes a protein crucial for biofilm formation of *Streptococcus pneumoniae* [84] and inhibits biofilm production in other bacteria [85].

Arnold et al. (1982) showed that the preincubation of *S. mutans* with human lactoferrin reduced glucose uptake and inhibited the synthesis of lactic acid, which indicated that lactoferrins can also affect glucose metabolism [59]. Furthermore, synergistic actions of lactoferrins with lysozymes (the major enzymatic components in the granules of polymorphonuclear lymphocytes) [64], bacteriophages [86], and antibiotics against different bacteria have been demonstrated [85]. By releasing LPS from the Gram-negative bacteria outer membrane, lactoferrins increase the permeability of the outer bacterial membrane and the susceptibility of Gram-negative bacteria to lysozymes, which have an important role in mammal defense mechanisms [64].

Furthermore, it has been demonstrated that lactoferrin also have anti-adhesive effects for some bacteria in upper respiratory mucosa and intestinal tract [87,88,89]. Lactoferrin selectively inactivate IgA1 protease and Hap adhesion in *Haemophilus influenzae* [88] and prevented the adhesion of enterotoxigenic *E. coli* to host cells [87,89].

As already mentioned, bovine and human lactoferrins contain glycosylation sites. Since glycosylation may play a role in the antimicrobial activity of proteins and peptides, a role of glycans in antimicrobial activity of lactoferrins was also proposed [36,37]. A study by Barboza et al. demonstrated that N-glycans from human lactoferrin inhibited the adhesion of *Salmonella enterica typhimurium*, *Salmonella enterica enteridis*, and *L. monocytogenes*, whereas N-glycans had no anti-adhesive effect on *E. coli* [90]. Furthermore, a study by Kautto et al. (2016) also demonstrated the anti-adhesive effect of human lactoferrin N-glycans for *Pseudomonas aeruginosa* to tears [91]. Another study revealed that sialylated glycans are responsible for the antimicrobial activity of lactoferrin, whereas neutral glycans had no effect [92].

### 2.1. Lactoferricins

The first peptide derived from the enzymatic hydrolysis of lactoferrin with pepsin was lactoferricin. The discovery that this lactoferricin peptide had better antibacterial activity than the native lactoferrin led to numerous studies on natural (obtained through enzymatic hydrolysis) and synthetic peptides. These studies have also provided greater understanding of the mechanism of antibacterial activity on the molecular level. Furthermore, in vitro and in vivo studies of gastric lactoferrin degradation have shown that many of these peptides, including lactoferricins, occur naturally in the body [93,94,95].

Lactoferricin was first isolated from bovine lactoferrin by Tomita et al. (1991) [58]. This was based on equal or stronger antibacterial activity upon digestion of bovine lactoferrin by exposure to acidic conditions (pH 2–3) and high temperatures (100 °C–120 °C) [96,97]. Then, following the enzymatic digestion of lactoferrins with pepsin, the resulting hydrolysates inhibited the growth of numerous Gram-positive and Gram-negative bacteria, including some bacteria that were resistant to lactoferrins [58]. The two best known lactoferricins are the bovine and human forms, although other lactoferricins have been isolated or synthesized based on the caprine, murine, and porcine lactoferrins [98,99]. As the most studied, this review focuses on bovine and human lactoferricin. These lactoferricins have antibacterial activities against a great variety of Gram-positive and Gram-negative bacteria (Appendix A). In general, the antibacterial activity bovine lactoferricin is greater than that for others species [57,98], and Gram-positive bacteria are generally more sensitive to bovine lactoferricin that Gram-negative bacteria [100,101].

The lactoferricins are cleaved from the N-terminal region of the lactoferrin molecule (Figure 1). These lactoferrin-derived peptides from different mammalian sources show differences in their amino-acid sequences, numbers of amino acids, and structures, all of which can influence their antimicrobial activities. Bovine lactoferricin consists of 25 amino acids that correspond to amino acids 17 to 41 from the N-terminus of bovine lactoferrin (Figure 3B) [57]. Bovine lactoferricin adopts a cyclic structure, which appears to be important in terms of its greater antibacterial activity, compared with its linear form [98]. This could be due to the stronger binding to the negatively charged membranes of the peptide form in comparison with the linear form and its deeper penetration into the membrane bilayer [102].

For human lactoferricin, the number of amino acids and the structure originally reported by Bellamy et al. (1992a) indicated 47 amino acids (1–47 of human lactoferrin) and included a region homologous to bovine lactoferricin. The sequence of human lactoferricin was reported as two subfragments connected by disulfide bonds between cysteine residues: the linear residues 1 to 11 and the cyclic residues 12 to 47 (Figure 3C) [57]. However, Hunter et al. (2005) later reported human lactoferricin as a 49-amino-acid peptide. They also proposed different human lactoferricin structures that are cyclic, although with a continuous polypeptide chain (Figure 3D) [70].

The crystallographic structure of lactoferricins has also been investigated. The secondary structure of bovine lactoferricin in aqueous solution defined by nuclear magnetic resonance revealed a slightly distorted antiparallel β-sheet [103]. This is markedly different from its X-ray structure in the intact parent bovine lactoferrin, which showed that the lactoferricin fragment (amino acids 12–29) forms an α-helix [17]. Hunter et al. (2005) studied the three-dimensional structure of human lactoferricin in aqueous solution using a membrane-mimetic solvent (dodecylphosphatidylcholine and sodium dodecyl sulfate). They showed that, in the region from Gln14 to Lys29, human lactoferricin in aqueous solution assumes a nascent helix in the form of a coiled conformation, while in the membrane mimetic solution, human lactoferricin showed helical content [70]. However, Chapple et al. (2004) demonstrated that the active parts of human lactoferrin and lactoferricin adopt a β-strand conformation rather than an α-helix in the presence of LPS [104]. Similarly, based on computer modeling of the structure of the N-terminus of human lactoferrin, Farnaud et al. (2004b) showed two β-strands separated by a strong turn rather than an α-helical structure [105]. These conformational differences might explain why lactoferricins have improved antimicrobial activities over their native lactoferrins. It also appears that the β-sheet structure of lactoferricins is better suited for the initial contacts with membranes and LPS than the α-helical structure of the native lactoferrins [103,105]. Furthermore, Pei et al. (2020) suggested that bovine lactoferricin shows a changed molecular structure according to ionic strength and hydrophobic effects, which improves its antibacterial activity [106]. The improved antibacterial activity of bovine lactoferricin (MIC 80 µM) over human lactoferricin (MIC 320 µM) has been attributed to the different distribution of the charge surrounding their hydrophobic cores [105].

Several studies have investigated the relationships between the structures of lactoferricins and lactoferricin-like peptides and their antibacterial activities. Even smaller fragments of lactoferricin sequences have been shown to have some antibacterial activity, such as the 15-amino-acid fragment (amino acids 17–31) [107,108,109,110,111]. It appears that the region of 11 amino acids from amino acids 20 to 30 in bovine lactoferricin is essential for the antibacterial activity, with this region showing similar to or greater antibacterial activity than bovine lactoferricin, although its hemolytic activity was lower [110,112,113]. With the introduction of a disulfide bond, the resulting cyclic version of this 11-amino-acid peptide showed greater general antimicrobial efficiency [113]. Conversely, the smaller 9-amino-acid (residues 20–28) and 6-amino-acid (amino acids 20–25) lactoferricin-derived peptides show no inhibitory activities against Gram-positive and Gram-negative bacteria [109,110,114].

To determine the active regions of bovine and human lactoferricins, several synthetic peptides were made for each. All bovine lactoferricins inhibited bacterial growth; however, only to modified human lactoferricins showed activities against the bacteria tested, while non-modified peptides that corresponding to different regions of native human lactoferricin showed no antimicrobial activities [109]. In contrast, the small 11-amino-acid peptide (21–31 of human lactoferrin) that is homologous to the helical and loop regions of human lactoferrin and lactoferricin, respectively, inhibited growth of *E. coli* and bound LPS [104]. This thus indicated that this region has an important role in the antibacterial activity of human lactoferrin and lactoferricin.

The charge and amino acid composition of lactoferricins also have important roles in their antimicrobial activities. Most natural antimicrobial peptides, such as lactoferricin, are cationic in nature, containing about +2 to +9 charge in them. This highly cationic nature allows peptide to interact with LPS in outer membrane of Gram-negative bacteria [115,116]. It has been shown that only lactoferricin-like peptides that contain a net positive charge of at least +5 have antimicrobial activities, with the most important cationic amino acid as Arg5, as its replacement with alanine resulted in decreased antibacterial activity [109,117].

Many natural antimicrobial peptides are rich in certain amino acids such as tryptophan, glycine, and histidine. Many of these peptides are known to have membrane permeability and intracellular targets. It seems that tryptophan residues play a role in promoting enhanced peptide-membrane interactions [116]. The tryptophan residues are also important for the antibacterial activity of bovine lactoferricin, as a replacement of either one of the two tryptophan residues (i.e., Trp6 or Trp8) led to severe loss of antibacterial activity [109,117]. Furthermore, Haug et al. (2001) indicated that the size, shape, and aromatic character of the tryptophan residues are the most important features for the antibacterial activities of bovine lactoferrin peptides [108]. The loop region of human lactoferricin contains only one tryptophan residue (Figure 3D), while the same region in bovine lactoferricin contains two tryptophan residues (Figure 3B), which might also be a reason for the reduced antibacterial activity of human lactoferricin. Substitution of all of the basic amino acids with glutamic acid and all of hydrophobic amino acids with alanine resulted in no antimicrobial activities for various lactoferricin-like peptides [112]. This thus showed the important role of these amino acids in the antibacterial mechanism of lactoferrin and lactoferricin activities. In contrast, it has been reported that cysteine disulfide bonds have no role in the antibacterial activity of bovine lactoferricin [106,110]. Hao et al. (2017) studied the antimicrobial activities of a 15-amino-acid bovine lactoferricin (amino acids 17–31) with its derivatives with different physicochemical properties [118]. Among the five peptides tested, the two designated as lactoferricin4 and lactoferricin5 showed the highest antibacterial activities. These two peptides had the highest α-helix content, the greatest number of hydrophobic amino acids, and a net charge of 5+. All of these properties appear to have contributed to the improved antibacterial activities of these two peptides in comparison with the others. To enhance the antibacterial activities or the spectrum of susceptible bacteria, several synthetic analogs of lactoferricins with different characteristic have also been made [119,120].

With the amino-acid sequence of lactoferricins devoid of the tyrosines, histidines, and aspartic acids that are involved in iron binding, their mechanism against bacteria is bactericidal [57]. As for lactoferrins, diverse mechanisms of this antibacterial activity of lactoferricins have also been demonstrated. Lactoferricins include a number of hydrophilic and positively charged amino acids that surround a hydrophobic surface (hydrophobic amino acids: Phe1, Cys3, Trp6, Trp8, Pro16, Ile18, and Cys20), which defines their amphipathic and highly cationic character [103]. According to the amphiphilic structure of lactoferricins in solution, they can interact with biological membranes and with anionic compounds in the bacterial outer membrane or cell wall, such as LPS or lipoteichoic acid, and cell wall teichoic acid. Umeyama et al. (2006) showed that bovine lactoferricin has greater affinity for acid phospholipids than for neutral ones. With dimyristoylphosphatidylglycerol representing the major component of the phospholipid bilayer of Gram-positive bacteria, bovine lactoferricin interacts specifically with the cell membrane of bacteria rather than the eukaryotic cell membrane [121]. Furthermore, the lactoferricin affinities for acid phospholipids are even greater than those of their native lactoferrins, which might also explain the improved antibacterial activities of lactoferricins in comparison with lactoferrins. Umeyama et al. (2006) also showed that bovine lactoferricin can form pores in acid phospholipid membranes. Diarra et al. (2003) reported that bovine lactoferricin causes deformation of the *Staphylococcus*
*aureus* cell wall, followed by cell lysis [122]. However, studies that have investigated the effects of bovine lactoferricin and a smaller 11-amino-acid fragment (amino acids 20–30) on membranes of Gram-negative bacteria indicated that the antibacterial mechanism of action was not via cell lysis [113,123]. Moreover, in these studies, Ulvatne et al. (2001b) demonstrated that bovine lactoferricin depolarizes the membrane of susceptible bacteria and induces fusion of negatively charged liposomes rather than causing bacterial lysis or major leakage from liposomes [123].

The absence of bacterial lysis with lactoferricins indicates that they can enter bacteria cells and target bacterial intracellular mechanisms. However, to achieve this, they need to first interact with components in the bacteria membrane or cell wall, so that the peptide can be translocated into the bacteria. It has been demonstrated that bovine and human lactoferricin bind to LPS in the outer membrane of Gram-negative bacteria, which results in the release of the LPS [61,104,109,124]. For bovine lactoferrin, the main binding site on LPS has been reported to be lipid A [68]; however, Farnaud et al. (2004a) suggested instead that lipid A is not the main binding site for bovine lactoferricin [109]. Based on their data, they presented a two-step mechanism where the positive amino acids of the cationic peptide first interact with the negative charges carried by LPS to cause disorganization of the structure of the outer membrane. This then allows the tryptophan residues to approach lipid A to establish hydrophobic interactions, which lead to further penetration of the outer membrane.

For Gram-positive bacteria, it is assumed that the binding site for lactoferricins are lipoteichoic acids and/or teichoic acids. The cell wall of Gram-positive bacteria is primarily composed of peptidoglycans. Lipoteichoic acids and teichoic acids are two anionic cell surface polymers that are located between these layers of peptidoglycans. Teichoic acids are covalently attached to peptidoglycans and are most commonly composed of disaccharide units to which are attached polyribitol phosphate or polyglycerol phosphate chains. Lipoteichoic acids are anchored to the head groups of the membrane lipids and are primarily composed of polyglycerol phosphate polymers that are often functionalized with D-alanine or a sugar moiety (Figure 2). Collectively, these polymers can account for over 60% of the mass of the cell wall of Gram-positive bacteria, which makes them major contributors to the envelope structure and function [107]. Vorland et al. (1999a) demonstrated that both lipoteichoic acids and teichoic acids can bind bovine lactoferricin, although teichoic acids bind bovine lactoferricin about 3–4-fold more efficiently than lipoteichoic acids [107]. Based on their data, they concluded that teichoic acids are the initial binding site for bovine lactoferricin.

An influence of lactoferricins on bacterial intracellular mechanisms has also been demonstrated. First, three types of lactoferricins (i.e., bovine lactoferricin 17–41, bovine lactoferricin 17–31, and D-bovine lactoferricin 17–31) were reported in the cytoplasm of both Gram positive (*S. aureus*) and Gram-negative (*E. coli*) bacteria. For *S. aureus*, the amounts of cytoplasmic bovine lactoferricin 17–41 were time-dependent and concentration-dependent and reached a maximum within 30 min [125]. Furthermore, Ulvatne et al. (2004) showed that low concentrations of bovine lactoferricin are insufficient to kill bacterial cells, with effects on the synthesis of DNA, RNA and proteins in Gram-positive and Gram-negative bacteria. In the Gram-positive bacterium *Bacillus subtilis*, bovine lactoferricin inhibited the synthesis of all bacterial macromolecules for at least 20 min, and thereafter, the synthesis of RNA increased. In the Gram-negative bacterium *E. coli*, bovine lactoferricin resulted in an initial decrease in DNA synthesis, parallel to an increase in RNA and protein synthesis, with profound filamentation of the *E. coli* seen. Bovine lactoferricin also inhibited bacterial protein synthesis in a concentration-dependent manner [101]. Similar data were obtained by Hao et al. (2017) for different derivatives of bovine lactoferricin 17–31. They demonstrated that two of these (i.e., lactoferricin4 and lactoferricin5) bound to genomic DNA of *S. aureus*, *Salmonella enteritidis* and *E. coli* and that lactoferricin4 also inhibited DNA, RNA, and protein synthesis in *S. aureus* [118]. Furthermore, Ho et al. (2012) demonstrated that bovine lactoferricin binds and inhibits phosphorylation of the response regulators BasR and CreB, and their cognate sensor kinases [126]. Yamazaki et al. (1997) reported for the first time that bovine lactoferricin can inhibit urease in *Helicobacter pylori* [127].

The antibacterial activities of lactoferricins are pH dependent and are reduced by various ions, such as Ca^2+^, Mg^2+^, Na^+^, and K^+^, with a similar trend also reported for lactoferrins [61,100]. This phenomenon has also been observed for other antimicrobial peptides, including bactenecins [128], polymyxin B [129], and the synthetic peptide TS [130]. The antibacterial activities of human lactoferrin and lactoferricin can be reduced not only by cations but also by anions, such as HEPES, PIPES, phosphate, citrate, and succinate [72,131].

### 2.2. Other Lactoferrin-Derived Peptides with Antimicrobial Activities

The discovery of lactoferricins led to several analyses on the hydrolysis of bovine lactoferrin, with the isolation of numerous new lactoferrin peptides with antibacterial activities against Gram-positive and Gram-negative bacteria, as listed in Appendix A. Bovine lactoferrin was most often used because of its greater accessibility and easier purification. Pepsin was most often used for enzyme hydrolysis, although the experimental conditions used in each study were often different to those used originally by Tomita et al. (1991) [58]. In this way, Dionysius et al. (1997) isolated three peptides from bovine lactoferrin, where peptides I and II showed antibacterial activities toward a number of pathogenic and food-spoilage microorganisms while peptide III showed lower activity [132]. Soon after, Recio and Visser (1999) used two ion-exchange chromatography methods for analysis of pepsin hydrolysates from bovine lactoferrin to isolate five peptides. Of these, their peptides 2 and 4 inhibited the growth of *Micrococcus flavus* [133]. Then, in 2016, Kim et al. demonstrated an inhibition of the growth of *Pseudomonas syringae* by a new peptide from bovine lactoferrin, again following enzymatic hydrolysis with pepsin [134]. Although this 10-amino-acid peptide originated from the N-terminus of bovine lactoferrin, it was positioned away from bovine lactoferricin (as for other peptides described above), as it was composed of amino acids 308 to 317.

Other enzymes have also been used for proteolytical cleavage of lactoferrin. When trypsin, papain, and enzymes from different bacteria were used for the cleavage, their hydrolysates generally showed lower antibacterial activities compared with the pepsin hydrolysates [58]. Then, when Lizzi et al. (2016) used trypsin for the hydrolysis of bovine lactoferrin, they showed that the whole bovine lactoferrin hydrolysate had the same antimicrobial activity against the Gram-positive and Gram-negative bacteria they tested [75]. However, the peptides of <5 kDa showed greater inhibition of bacterial growth than the native lactoferrin, while the peptides of >5 kDa showed no inhibition, which indicated that only the small peptides retained antibacterial activities. Rastogi et al. (2014) also performed enzymatic hydrolysis of bovine lactoferrin with trypsin, and they reported on the isolation of three peptides. In comparison with the native lactoferrin, all of these showed greater activities toward Gram-negative bacteria compared with Gram-positive bacteria [135]. Hoek et al. (1997) used recombinant chymosin for the enzymatic cleavage of bovine lactoferrin and produced four peptides that had greater antimicrobial activities than that for the native lactoferrin, one of which had the bovine lactoferricin amino-acid sequence [136].

As seen for lactoferricin, all of these peptides obtained by enzymatic hydrolysis were cationic and located in the N-terminus of the bovine lactoferrin molecule (Figure 4). Additionally, for lactoferricin, none of these peptides contained any of the amino acids involved in iron binding.

### 2.3. Lactoferrampin

Antimicrobial peptides have been known for many decades, and they are now being more extensively studied because of their huge potent as natural antibiotics. These host defense peptides, as they are known, have some common features, which include being short (10–50 amino acids), positively charged (generally +2 to +9), and hydrophobic (≥30% hydrophobic residues) [138].

Based on such common features of antimicrobial peptides, a new peptide was synthesized that is known as lactoferrampin. The amino-acid sequence of this synthetic peptide corresponds to amino acids 268 to 284 of bovine lactoferrin, which is in the N-1 domain (Figure 4). Lactoferrampin has shown a broad spectrum of antibacterial activity, although some bacteria are resistant to this peptide, including *Porphyromonas*
*gingivalis*, *Ac**tinomyces*
*naeslundii*, *S. mutans*, and *Streptococcus sanguis*. In addition, lactoferrampin has no hemolytic activity at the antimicrobial working concentration [139]. Based on the common features of antimicrobial peptides, it became possible to isolate lactoferrampin and other lactoferrampin-like peptides from bovine lactoferrin using enzymatic actions, as their cleavage sites were well known. Using the ‘*PeptideCutter*’ option of the ExPASy Proteomics Server, Bolscher et al. (2006) predicted cleavage sites of bovine lactoferrin with site-specific endoproteinases such as ArgC (clostripain), AgrN, and ArgC/AgrN for the release of lactoferrampin and three other peptide fragments f(259–284), f(265–296), and f(265–284) [140]. All four of these showed antibacterial activities against *E. coli*.

Based on this region for bovine lactoferrampin, human lactoferrampin was also synthesized, which corresponds to amino acids 269–285 [141]. This human lactoferrampin and the same amino acids with the addition of an N-terminal helix cap, where the corresponding sequence in human lactoferrin is DAI (cap human lactoferrampin), showed no effects on *E. coli* and *S. sanguis*. Only when Asp17 was exchanged for asparagine in cap human lactoferrampin (i.e., cap-LFampH-K D17N) or when a lysine residue was added to the C-terminus, was the inhibition of bacterial growth obtained.

### 2.4. Lactoferrin-Chimera

As indicated in Figure 4, the chimerization of lactoferricin 17–30 and lactoferrampin 265–284 produced a new peptide composed of 35 amino acids and designated as lactoferrin-chimera. Combining these two antimicrobial peptides into one molecule resulted in greater antibacterial activity against several Gram-positive and Gram-negative bacteria (Table 1) than for lactoferricin 17–30 or lactoferrampin 265–284 alone and for their mixture [84,140,142,143]]. The synergistic effects of lactoferrin-chimera with different antibiotics against multi-drug resistant *Vibrio parahaemolyticus* has also been reported [142]. It was demonstrated that lactoferrin-chimera also inhibited the growth of some strains of *Burkholderia pseudomallei* that were even resistant to the antibiotic of choice, ceftazidime [144].

When compared with lactoferricin and lactoferrampin, lactoferrin-chimera showed bacterial inhibition at lower concentrations, over shorter incubation times (maximum activity within 15 min), and with less sensitivity to ionic strength (i.e., NaCl 50/100 mM). It has been shown that all three of these peptides interact with negatively charged dimyristoylphosphatidylglycerol liposomes when used as a model for bacterial membranes, where the interaction with lactoferrin-chimera was the strongest, while that for lactoferricin was the weakest [84,137,142,145]. The mechanism behind this enhanced antibacterial activity of lactoferrin-chimera might arise from its structure. Circular dichroism spectroscopy here revealed an α-helix for lactoferrin-chimera and lactoferrampin 265–284, and a β-turn for lactoferricin 17–30 [137]. As this α-helix of lactoferrin-chimera mimics the spatial arrangement in the native lactoferrin, where the antibacterial activity is weaker than for lactoferricin and lactoferrampin, it is more likely that the reason for this enhanced antibacterial activity lies in the lactoferrin-chimera mechanism. Its net positive charge of 12+ allows for binding to bacterial membranes, and consequently, destabilization and permeabilization of the bacterial membrane [84,137,142,145]. Membrane penetration or translocation with *S. pneumoniae* was also seen for lactoferricin 17–30 and lactoferrampin [84,136,142].

## 3. Antiviral Activities of Lactoferrin and Lactoferrin-Derived Peptides

As well as their antibacterial activities, lactoferrins also have broad activity against capsulated and naked viruses that cause diseases in both humans and animals, as indicated in Appendix A. Antiviral activities against some viruses have also been seen for lactoferricins and other lactoferrin-derived peptides; however, the antiviral activities of lactoferricins are often weaker than that of their native protein [146].

The mechanisms of the antiviral activities of lactoferrins, lactoferricins, and lactoferrin-derived peptides include the prevention of virus-induced apoptosis of cells [147,148] and the prevention of virus entry into cells by binding to virus envelope proteins or to the virus receptors on the cells [149,150,151,152,153,154,155,156,157,158,159,160,161,162,163,164,165,166,167,168,169,170,171] or by influencing other intracellular virus mechanisms [147,169,172,173]. These mechanisms are more extensively described below for the individual enveloped and naked viruses.

### 3.1. Influenza Virus

Influenza viruses are enveloped RNA viruses that cause respiratory infections in humans and animals and are responsible for high morbidity and mortality, especially among people with immunodeficiency associated with aging or underlying predisposing conditions. Inhibitory effects of lactoferrin towards the influenza A viruses H1N1, H3N2, and H5N1 have been reported [147,149,174]. Lactoferrin shows different mechanisms for its antiviral activities against these influenza A viruses. Pietrantoni et al. (2010) demonstrated that bovine lactoferrin prevents virus spread by preventing apoptosis of the infected cells, potentially through the prevention of caspase 3 activity [147]. Caspase 3 is a cysteinyl protease that has one of the crucial roles in the regulation of apoptosis [175]. Their study also revealed that bovine lactoferrin efficiently blocked nuclear export of viral ribonucleoproteins, thus preventing viral assembly [147].

Furthermore, it has been shown that lactoferrin interacts with influenza A virus hemagglutinin (as the HA_2_ segment; see below) and prevents infection by different H1 and H3 viral subtypes [149,157]. Hemagglutinin is a glycoprotein that is expressed on the viral envelope along with neuraminidase, and it has a crucial role in virus binding and entry into host cells. Hemagglutinin is homotrimeric, and each monomer is composed of two polypeptide segments, designated as HA_1_ and HA_2_ [176]. The HA_1_ segments mediate hemagglutinin attachment to the host cell surface through interactions with sialic acid side chains of receptors on the host cell surface. After a conformation change of the HA_2_ segment caused by low pH, the fusion peptide is exposed. This fusion peptide is translocated to the endosomal membrane, thereby mediating fusion of the viral envelope with the membranes [149,177]. It appears that the antiviral activity of lactoferrin against influenza A viruses is entirely mediated by the C-lobe of lactoferrin, with no inhibition shown by the N-lobe. The C-lobe also shows greater antiviral activity against influenza viruses than the native bovine lactoferrin. Trypsin digestion of the C-lobe resulted in three fragments that inhibited both hemagglutination and virus infection, as peptide 1 (amino acids 418–429), peptide 2 (amino acids 506–522), and a modified sequence of peptide 3 (amino acids 552–563). Each of these three peptides had greater inhibitory hemagglutinin activity than the C-lobe, with the most potent antiviral activity seen for peptide 1 [149]. Based on this, peptide 1 was further investigated to determine the region or structural requirements that determined this bovine lactoferrin C-lobe interaction with hemagglutinin. Here, Scala et al. (2017) produced several synthetic peptides; however, only three of these peptides showed inhibitory effects against both the H1N1 and H3N2 strains: peptide 14 (amino acids 426–429), peptide 15 (amino acids 422–425), and peptide 17 (amino acids 418–421). Peptide 17 also showed greater antiviral activity than peptide 1 [178].

### 3.2. Hepatitis C Virus and Hepatitis B Virus

Antiviral activities of bovine lactoferrin and human lactoferrin against the hepatitis *C. virus* (HCV), as the causative agent of liver cirrhosis and hepatocellular carcinoma [179], have been reported [178]. Using cultured human hepatocytes (PH5CH8 cells) that are susceptible to HCV replication [178], they demonstrated that bovine and human lactoferrin inhibited infection of these cells by a rapid (<1 min) and direct interaction with HCV, rather than with the cultured cells; this prevented entry of HCV into these cells [178]. Further studies from two independent laboratories revealed that both bovine and human lactoferrin bound equally well to two glycoproteins in the surface envelope of HCV, E1, and E2 [180,181]. In particular, E2 glycoprotein was important for virus entry into the host cell, while E1 was believed to be the fusogen (i.e., glycoprotein that facilitates cell fusion with the virus) [179]. However, when lactoferricin 17–42 was tested, no anti-HCV activity was seen. To identify the region responsible for bovine lactoferrin binding to E2 glycoprotein, Nozaki et al. (2003) performed fragmentation of a human lactoferrin cDNA clone from *E. coli*. This provided a peptide, designated as C-s3 (amino acids 600–692), that was involved in direct interactions with E2 glycoprotein and prevented entry of HCV into cells. Furthermore, fragment C-s3 showed binding activity similar to that of human lactoferrin. This C-s3 peptide was further fragmented, whereby the 33 amino acid sequence of human lactoferrin (amino acids 600–632) was identified as the main peptide that is directly involved in binding to E2 glycoprotein. It appears that a cysteine residue (Cys628) has an important role in the binding of the C-s3 600–632 fragment to E2 glycoprotein, as its substitution with alanine completely abolished this binding. Additionally, when the bovine lactoferrin 597–629 sequence (that corresponds to human C-s3 fragment 600–632) was tested, it was shown to bind equally well to E2 glycoprotein [151]. Bovine lactoferrin also showed anti-HCV activity in MT-2C cells (a human T-cell cell line), which indicated that bovine lactoferrin can prevent HCV infection of hepatocytes and lymphocytes [151]. When bovine and human lactoferrins were preincubated with PH5CH8 cells, this also prevented infection of these cells with hepatitis B virus (HBV). The preincubation of HBV with bovine or human lactoferrin had no inhibitory effects on HBV infection, which indicated that the mechanism of antiviral activity is the same for HCV and HBV [151].

### 3.3. Herpes Simplex Virus and Human Cytomegalovirus

Antiviral activities of bovine and human lactoferrins and their bioactive lactoferricin peptides against human Herpes simplex virus 1 (HSV-1) and HSV-2 have been reported. Anti-HSV-1 and anti-HSV-2 activities have also been shown for bovine and human lactoferricins. Native lactoferrins showed greater antiviral activities against HSV-1 and HSV-2 in comparison with lactoferricins, although bovine lactoferricin was more potent that human lactoferricin [146].

To determine the region responsible for these antiviral activities, bovine lactoferrin was digested with trypsin. Among the 31 fractions obtained, only three showed antiviral activities: two large fragments, designated as fractions 30 (amino acids 345–689) and 28 (amino acids 1–280), and one low molecular weight fragment, as fraction 19. The peptide from fraction 30 (from the N-lobe) was more potent than the peptide from fraction 28 (from the C-lobe); however, the larger peptides showed lower antiviral activities compared with the native bovine lactoferrin. Fraction 19 contained two small peptides that were then synthesized chemically for evaluation: peptides with sequences f(222–230) and f(264–269), although when these were tested separately, neither of them showed any antiviral activities against HSV-1 [182].

The mechanism of virus entry into host cells is mediated by virus attachment to glycosaminoglycans as mostly heparan sulfate, in cell surface receptor proteoglycans, via the viral glycoproteins gB and gC [183]. Early studies indicated that the mechanism of lactoferrin antiviral activity relates to the early stages of virus infection to prevent virus attachment or penetration into the host cell [153,154,155]. Furthermore human lactoferrin can bind to heparan sulfate [184]. Later studies demonstrated that the lactoferrin antiviral activity against HSVs is diverse. One part of the lactoferrin antiviral activity is to prevent entry of HCV-1 into host cells by binding to heparan sulfate and/or chondroitin sulfate glycosaminoglycans in the cell membranes to occupy the initial places of virus binding. It has also been shown that the antiviral activities of lactoferrins against HSV-1 depended on gC (the attachment protein that mediates virus binding to heparan sulfate) and on heparan sulphate at the cell surface. This cell surface heparan sulfate is also important for lactoferricin inhibition of virus entry into the host cells [152,156]. Bovine lactoferrin also targets the HSV-1 entry process and interferes with viral trafficking towards the nucleus, which thus indicates another aspect of the lactoferrins anti-HSV activities. By adding bovine lactoferrin immediately after the viral adsorption step, about 40% of the virus production was inhibited; however, this process was only observed within 40 min [173]. The addition of bovine lactoferrin after 40 min of infection results in no inhibition of HSV-1 [156,173]. Immunoflourescence assays showed that lactoferrin was primarily on the cell surface, while bovine lactoferricin was also distributed intracellulary [152,173]. Although some of the bovine lactoferrin was also seen at the plasma membrane and in the cytoplasm of the cells, this indicated that bovine lactoferrin can penetrate into the cells [152,173]. Furthermore, it has been shown that bovine lactoferrin and lactoferricin interfere with viral trafficking toward the nucleus and with HSV-1 replication [172]. It appears that intracellulary, bovine lactoferrin targets ICP-5 (the viral capsid protein) and VP-16 (the viral tegument protein), thus preventing viral assembly and viral translocation to the nucleus [173]. Bovine lactoferrin and lactoferricin also target the cell-to-cell spread of HSV to prevent plaque formation [152,173]. Iron binding had no effect on the antiviral activities of lactoferrins against HCV-1 and HCV-2 [154,155]. (Hasegawa et al., 1994; Marchetti et al., 1996).

Human Cytomegalovirus (HCMV) is a member of the Herpesviridae family, and it can cause mononucleosis syndrome and hepatitis in immunocompetent hosts. Antiviral activities of lactoferrins against HCMV were reported in the mid-1990s by Hasegawa et al. [154] and Harmsen et al. [159]. Similar to that for HCV, with HCMV, human, bovine, and goat lactoferrins prevented virus adsorption and/or penetration into the host cells to influence early events in virus infection [154,158,159]. Among these three lactoferrins, bovine lactoferrin was the most potent. These differences in antiviral activities might be due to some small structural differences between these molecules. Cyclic lactoferricin H also showed anti-HCMV activity; however, this activity was weaker by a factor of seven compared to the native lactoferricin. Furthermore, linear lactoferricin B and lactoferricin H showed no inhibitory effects, with some inhibition seen for lactoferricin G [158].

### 3.4. Human Immunodeficiency Virus

Lactoferrins have antiviral activities against human Immunodeficiency virus type 1 (HIV-1) and HIV-2 [159,160,161,162]. The activities of bovine lactoferrin were more potent than for human lactoferrin [159,160]. It appears that a globular structure and a net negative charge are both required for anti-HIV-1 activity [159], as both bovine and human lactoferrins showed greater antiviral activities against HIV-1 than bovine lactoferricin [160]. Additionally, derivatization of lactoferrins with succinic anhydride produced the negatively charged succinylated lactoferrins, which showed increased anti-HIV-1 activity [159,162]. Puddu et al. (1998) showed that metal saturation of bovine lactoferrin improved anti-HIV activity [161]. The precise mechanism of the anti-HIV activities of lactoferrins is not known; however, several studies have indicated that lactoferrins interfere with virus replication, probably at the level of virus–cell fusion and/or binding to thus prevent virus entry into host cells [159,160,161,162]. Lactoferrins also bind strongly to the V3 loop of the gp120 virus envelope protein [162]. Human and bovine lactoferricins and the two synthetic peptides lactoferricin 1–11 and lactoferrampin were investigated for their effects on three HIV-1 enzymes that are important in the HIV life-cycle: HIV-1 reverse transcriptase, protease, and integrase. This study showed that lactoferricin H, lactoferricin B, and lactoferrampin inhibited HIV-1 reverse transcriptase, with the first of these being more potent. Furthermore, lactoferrampin strongly inhibited HIV-1 integrase, while all of the peptides tested showed only slight inhibitory effects on HIV-1 protease [185].

### 3.5. Coronaviruses

Lactoferrin antiviral, immunomodulatory, and anti-inflammatory activity led to several hypothesis of lactoferrins as potential drugs against the recent SARS-COVID-19 pandemic caused by Severe acute respiratory syndrome coronavirus 2 (SARS-CoV 2) [46,47,48,49,50]. Coronaviruses are enveloped RNA viruses that generally cause respiratory and/or intestinal infections. Their entry into host cell is mediated through the interaction of coronavirus spike (S) proteins with angiotensin-converting enzyme 2 (ACE-2), a host cell receptor. Coronavirus spike protein is divided into two domains (S1 and S2). The S1 domain is responsible for receptor binding, e.g., binding to ACE-2, while the S2 domain is responsible for membrane fusion [186].

To our knowledge, no interaction of lactoferrins with ACE-2 receptors is reported; however, binding to heparan sulfate [184] and preventing virus entry into host cell have been demonstrated in a case of HSV. It has been shown that murine coronavirus [187] and human coronaviruses HCoV-NL63 [188] require heparan sulfate for virus attachment and/or entry into host cells. Furthermore, Lang et al. (2011) demonstrated that lactoferrin inhibits SARS-CoV pseudotype infections in a dose-dependent manner. They presumed that heparan sulfate is required for virus attachment, although an alternative inhibition of coronaviruses through binding to spike protein has also been proposed [189].

An observational study on 75 patients performed by Serrano et al. (2020) where liposomal lactoferrin was used, showed that lactoferrin can potentially be used for the prevention and treatment of SARS-CoV 2 [190]. Furthermore, studies by different research groups demonstrated that lactoferrins boost antiviral immune response and exerts in vitro antiviral activity against SARS-CoV 2 infection in a dose-dependent manner [191,192,193,194]. Multiple modes of action have been indicated, which includes blockage of the virus by binding to viral spike glycoprotein [192], blockage of virus attachment to cellular host cell receptors (e.g., heparan sulfate) [192,193,194] or enhancement of interferon responses [194]. A study by Hu et al. (2021) showed that bovine lactoferrin is more potent in SARS-CoV 2 inhibition than human lactoferrins, with 50% effective concentration (EC50) values ranging from 11.2 to 37.9 μg/mL for bovine lactoferrin and EC50 values ranging from 35.7 to 117.9 μg/mL for human lactoferrin [193]. Antiviral activity against SARS-CoV 2 was also observed for the holo-form of lactoferrin [194].

### 3.6. Adenoviruses

The antiviral activities of lactoferrins against naked viruses have also been investigated. Bovine lactoferrin, human lactoferrin, and bovine lactoferricin were shown to have anti-adenovirus activities [163,164]. Here, lactoferrin acted through two different antiviral mechanisms: binding to the cell membrane glycosaminoglycans to prevent adenovirus attachment to its cell receptors and direct binding to adenovirus particles (to polypeptides III and IIIa) to influence viral replication [163,164,165]. When the N-lobe and C-lobe were tested separately, only the N-lobe showed cytopathic effects against adenovirus [164]. On the other hand, different activities have also been described for lactoferrins that are in contrast with these anti-adenovirus activities. Human lactoferrin was shown to promote binding of adenovirus to epithelial cells in a dose-dependent manner, as also for increased infection of epithelial cells by adenoviruses [195]. Furthermore, Adams et al. (2009) demonstrated that high levels of lactoferrin at mucosal sites can facilitate adenovirus (serotype 5) attachment and can enhance infection of dendritic cells [196]. Human lactoferrin also directly binds to adenovirus-C5, -D26, and -B35, with affinities in the micromolar range and increases human adenovirus uptake by mononuclear phagocytes [197].

### 3.7. Human Papilloma Virus

Bovine and human lactoferrins and their lactoferricins were also tested for antiviral activities against two types of human Papillomavirus (HPV): α-papillomavirus HPV-16 and β-papillomavirus HPV-5. Using immortalized human keratinocytes (HaCaT cells) and cells from cervical cancer mucosa (C33A cells), both bovine and human lactoferrins inhibited their infection by HPV-16 and HPV-5, with the antiviral activity of bovine lactoferrin greater than for human lactoferrin [166,167]. When three lactoferricins were tested, two of them, the linear bovine lactoferricin 17–31 and human lactoferricin 1–49, inhibited both HPV-16 and HPV-5, while the third, bovine lactoferricin 17–42, showed antiviral activity against HVP-5. These inhibitory effects were greatest for linear bovine lactoferricin 17–31. However, for inhibition of viral binding, only the circular lactoferricins showed activities [167].

### 3.8. Rotaviruses

Antiviral activities of lactoferrins against non-enveloped viruses were first reported for rotavirus. Superti et al. (1997) demonstrated that bovine lactoferrin prevented rotavirus hemagglutination and viral binding to susceptible cells by binding to the viral particles [170]. Metal binding (i.e., Fe, Zn, and Mn) did not interfere with these antiviral activities, while removal of sialic acid enhanced the anti-rotavirus activity of bovine lactoferrin. To clarify the mechanism of the anti-rotavirus effects of bovine lactoferrin, Superti et al. (2001) digested bovine lactoferrin with trypsin. Of all of the peptides they obtained, only two showed anti-rotavirus activities: a large peptide (amino acids 86–258) and a small peptide (amino acids 324–329) [198].

### 3.9. Echoviruses

Lactoferrins can prevent viral-induced apoptosis by enveloped viruses, such as influenza virus A; furthermore, for bovine lactoferrin, this was also demonstrated for a naked virus known as echovirus [148]. Furthermore, Pietrantoni et al. (2006) indicated that bovine lactoferrin affected not only virus adsorption and binding to target cells but also the later stages of viral infections, as inhibition of echovirus also persisted after the virus had bound to its cell membrane receptors. When bovine lactoferrin tryptic hydrolysates were tested for antiviral activities here, only the N-lobe and bovine lactoferricin inhibited echovirus, with the C-lobe inactive. Moreover, the N-lobe showed the same anti-echoviral activity as the entire lactoferrin, and it also maintained activity after virus attachment to the host cells; instead, lactoferricin only acted on the early stages of viral infection, whereby the bovine lactoferricin alone prevented viral attachment to the target cells [169].

## 4. Antifungal Activities of Lactoferrins and Lactoferrin-Derived Peptides

Antifungal activity for human lactoferrin was first reported by Kirkpatrick et al. (1971), where inhibition of growth was demonstrated against the yeast *Candida albicans* [199]. Human lactoferrin also inhibited the growth of *Candida krusei*, to a greater extent than seen for *C. albicans*. The inhibition of each of these was dose dependent [200,201], while iron saturation resulted in the loss of this antifungal activity of human lactoferrin [199,200,201]. Soon after the first isolation of bovine lactoferricin, its antifungal activity was extensively studied (Appendix A), including for fungi that cause dermatophytosis. Using [^14^C]-labeled bovine lactoferricin, its direct binding was shown [202], along with its potent disruptive effects on *C. albicans* cell membranes [203]. Furthermore, for pathogenic fungi, bovine lactoferricin inhibited the uptake of [^3^H]-glucose in *Trichophyton rubrum* and caused substantial changes to the ultrastructure of *Trichophyton mentagrophytes*, which included dense aggregation of the cytoplasmic materials [204]. The anti-*Candida* activity of bovine and human lactoferricins can be affected by different pHs, temperatures, and ions (i.e., phosphate, bicarbonate, Ca^2+^, and Mg^2+^) [201,202]. Indeed, bovine lactoferricin binding to *C. albicans* was reduced by the addition of the divalent cations Ca^2+^ and Mg^2+^,and was pH dependent, as also seen for its antimicrobial activity [202].

To further determine the antifungal activities of bovine lactoferricin, several lactoferricin-like peptides were tested by Ueta et al. (2001), using synthetic bovine lactoferricin and four of its derived peptides. Among these peptides, peptide 2 (amino acids 17–26) showed the greatest suppression of multiplication of *Candida* cells, while the other peptides showed only weak activities [205]. Munoz and Marcos (2006) tested two bovine lactoferricin-derived peptides, lactoferricin 17–31 and lactoferricin 20–25, against different bacteria and fungi that are causative agents for plant diseases [114], while van der Kraan et al. (2004) tested the anti-*Candida* activities of bovine lactoferricin fragment 17–30 [139]. These data showed that bovine lactoferricin (20–25) was less active than the more extended bovine lactoferricin (17–31), with the exception of *Botrytis cinerea*, where very similar activities were seen, and that filamentous fungi were more susceptible than bacteria or yeast [114]. Peptide 2 of Ueta et al. (2001) also did not bind iron, which indicated that its mechanism of anti-*Candida* activity was unrelated to depriving these yeast of this nutrient [205]. Three peptides obtained by tryptic digestion of bovine lactoferrin (i.e., 21LF, 38LF, and 45LF) showed lower antibacterial activities than the native protein; however, their antifungal activities were greater than that of lactoferrin [135]. Furthermore, the human lactoferricin 1–11 synthetic peptide inhibited biofilm formation by *C. albicans* mainly at the early stages, showing interference with the cell density of the biofilm and the metabolic activity [206].

*Candida albicans* has also shown high susceptibility to synthetic lactoferrampin 268–284 and other lactoferrampin-like peptides [139,140,207]. To define the antimicrobial region of bovine lactoferrampin, van der Kraan et al. (2005) synthesized a series of lactoferrampin peptides. They concluded that the positively charged amino acids of the C-terminal of lactoferrampin 265–284 were crucial for its candidacidal activity, while the N-terminal part was essential for activity because it facilitated helix formation [207]. Interestingly human lactoferrampin showed no inhibitory effects against *C. albicans* unless a lysine residue was added to the C-terminus of molecule or a negatively charged aspartic acid was mutated to asparagine [141].

## 5. Antiparasitic Activities of Lactoferrins and Lactoferrin-Derived Peptides

Lactoferrin and lactoferrin-derived peptides also exert antiparasitic activity against different protozoa and small parasites. Lactoferrin inhibited the in vitro growth of *Babesia caballi* and *Babesia equi*; however, the inhibitory effect was greater for *B. caballi* and only occurred in the presence of apo form of lactoferrin [208]. Since many microorganisms, including parasites, require iron for growth and development, iron binding proteins lactoferrin can contribute to host defense against parasites by sequestrating this important nutrient from microorganisms [209]. Furthermore, human and bovine lactoferrin ant lactoferrin-derived peptides (including lactoferricin, lactoferampin, and Lf-chimera) also showed antiparasitic activity against *Giardia lamblia* [210,211] and *Giardia intestinalis* [212]. The most gardicidal effect was observed for bovine lactoferrin-derived peptides (50% lethal dose (LD_50s_) of 8 µg/mL) followed by human-derived peptides, bovine lactoferrin (LD_50s_ of 1.2 mg/mL), and human lactoferrin (LD_50s_ of 1.5 mg/mL), indicating that bovine lactoferrin is more potent that human lactoferrin [210]. Furthermore, it has been shown that lactoferrins and lactoferrin-derived peptides bind on the surface of *G. lamblia* [211,213]. The gardicidal effect of lactoferrin and lactoferrin-derived peptides was also observed at low concentrations, where they caused dilation of the endoplasmic reticulum (ER) membranes, expansion of the nuclear membrane, and plasma membrane protrusions [211], although high concentrations cause severe morphological changes or even induce programmed cell death [212,213]. Lactoferrin also exhibited antiparasitic activity against *Cryptosporidium parvum* sporozoites but had no significant effect on oocysts viability or parasite intracellular development [214].

Some protozoan parasites such as Trichomonas, Giardia, and Entamoeba require a high extracellular iron concentration for their growth [215,216] and have therefore adapted to acquire extracellular iron from other sources such as host iron-binding proteins such as transferrin and lactoferin. It is well known that *Trypanosoma brucei* can sequestrate iron from transferrin by binding through a surface receptor [217]. Iron uptake from transferrin and lactoferrin has also been demonstrated for *Trichomonas vaginalis* [216], *Trichomonas foetus* [218], and *Leishmania chagasi* [219]; however, in the last case, other possible mechanism of iron sequestrating have been proposed [220]. Binding sites for lactoferrin has also been demonstrated in *T. brucei* [221] and *Toxoplasma gondii* [222,223,224]; however, in *T. gondii,* binding sites were specific for lactoferrin since the absence of transferrin binding was observed [224]. It is possible that protozoa binding sites for lactoferrin could have roles in iron acquisition; however, this is yet unclear since iron saturation had no impact on binding pattern [221,222]. Furthermore, studies by Tanaka et al. and Dzitoko et al. showed that lactoferrins did not prevent parasite penetration into host cell or had direct cytotoxic impact on *T. gondii* viability [225,226,227]; however, the inhibition of protozoa multiplication by lactoferrin was demonstrated [227]. Furthermore, when the lactoferrin-derived peptide lactoferricin was applied, a reduced viability, cyst formation in mouse brains, infectivity of sporozoites, and decreased penetration activity by *T. gondii* was observed [225,228,229]. Reduced infectivity of sporozoites was also observed in a case of *Eimeria stiedai* [229].

## 6. Conclusions

This review set out to clarify some aspects behind the mechanisms of lactoferrin and lactoferricin antimicrobial activities. This has revealed that lactoferrin and its derived peptides have a broad spectrum of antimicrobial activities that are closely connected to the protein or peptide amino-acid compositions, and their structures and conformations. Interestingly, the N-terminal-derived peptides showed greater antibacterial activities against bacteria and fungi, while for their antiviral activities, it appears that the whole protein is necessary. The information provided in this review provides us with a better understanding of the antimicrobial mechanics of these activities at the molecular level, which can now be applied for the production of novel antimicrobial peptides.

## Figures and Tables

**Figure 1 ijms-22-11264-f001:**
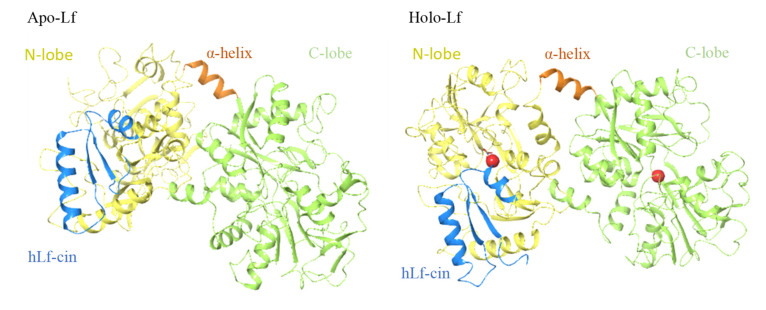
Structure of human apo-lactoferrin (**left**) and holo-lactoferrin (**right**). Yellow, N-lobes (amino acids 1–333); green, C-lobes (amino acids 345–691); orange, short joining α-helix (amino acids 334–344); blue, human lactoferricin cleavage peptide (amino acids 1–49).

**Figure 2 ijms-22-11264-f002:**
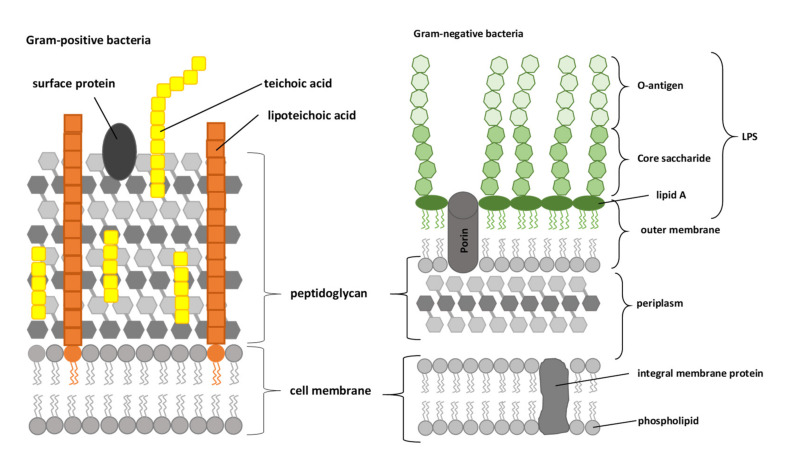
Compositions of the Gram-positive and Gram-negative bacterial cell envelopes. LPS, lipopolysaccharide, composed of O-antigen, core saccharide, and lipid A (adapted from [67]).

**Figure 3 ijms-22-11264-f003:**
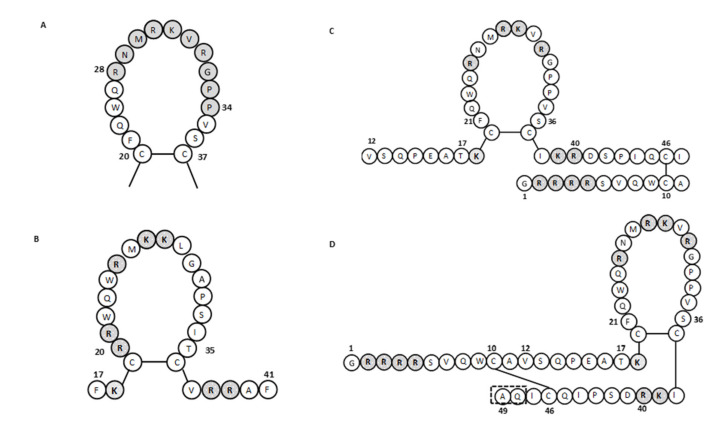
The amino acid sequence of lactoferrins and lactoferricins. (**A**) Loop region in the N-terminal lobe of human lactoferrin. Gray shading, amino acids involved in bacterial binding. (**B**–**D**) Bovine lactoferricin (**B**) and human lactoferricin, originally defined with 47 amino acids (**C**) and later with 49 amino acids (**D**). Gray shading, positively charged amino acids; dashed box in (**D**), two additional amino acids over (**C**) (adapted from [57,65,70]).

**Figure 4 ijms-22-11264-f004:**
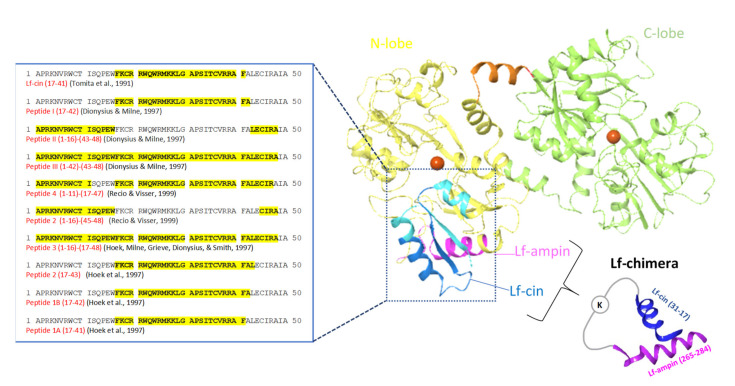
Structure of the bovine lactoferrins. Blue ribbon, antibacterial domain of bovine lactoferrin, from where the lactoferrin-derived peptides with antibacterial activities were obtained, with their amino-acid sequences given on the left. Darker blue ribbon, bovine lactoferricin (Lf-cin); violet ribbon, lactoferrampin (Lf-ampin; amino acids 268–284). Chimerization of lactoferricin 17–30 and lactoferrampin 265–284 produced lactoferrin (Lf)-chimera, where these two peptides were coupled through the two amino groups of a lysine (adapted from [137]).

**Table 1 ijms-22-11264-t001:** Mammalian species from where lactoferrins have been identified and isolated.

Order	Species	Source of Lactoferrin Isolation	Reference
Primates	Human	Colostrum, milk, tears, nasal/bronchial secretions, saliva, bile/pancreatic secretions (i.e., gastric/intestinal fluids), urine, seminal/vaginal fluids, granules of neutrophils	[4,5,7,9]
	Rhesus monkey	Milk	[10]
	Patas monkey, macaque, baboon, orangutan	Granules of neutrophils	[7]
Carnivores	Dog, bear, domestic cat, tiger, jaguar, cougar, meerkat, otter, tayra, palm civet	Granules of neutrophils	[6,7]
Rodents	Rat, hamster, aguti	Granules of neutrophils	[7]
	Mouse, guinea pig	Milk, granules of neutrophils	[4,7]
Lagomorpha	Rabbit	Granules of neutrophils	[7,11]
Artiodactyla	Sheep, buffalo, alpaca, camel	Milk	[9,12,13]
	Deer	Granules of neutrophils	[4,7]
	Cow, goat, pig	Milk, granules of neutrophils	[7]
Perissodactyla	Horse	Milk, granules of neutrophils	[4,7]
Proboscidea	Elephant (Asian and African)	Milk	[9,14]
Didelphimorphia	Opossum	Granules of neutrophils	[7]
Cingulata	Armadillo	Granules of neutrophils	[7]

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
