# Peer review of "Diverse Mechanisms of Antimicrobial Activities of Lactoferrins, Lactoferricins, and Other Lactoferrin-Derived Peptides"

_ijms, 2021, doi:10.3390/ijms222011264_

Round 1

Reviewer 1 Report

This is a well written, updated manuscript on microbial actions of lactoferrin and its peptides. It should be however, supplemented by other missing information, for example on indirect antibacterial activities  of lactoferrin and its synergistic actions with antibiotics or bacteriophages. In addition, the authors did not mention about anti-parasitic action of lactoferrin (Giardia lamblia).  Suggested literature below.

Hwang S-A, et al. Oral recombinant human or mouse lactoferrin reduces Mycobacterium tuberculosis TDM induced granulomatous lung pathology Biochem Cell Biol. 2017 Feb;95(1):148-154.

Nguyen TKT, et al. Lactoferrin reduces mycobacterial M1-type inflammation induced with trehalose 6,6'-dimycolate and facilitates the entry of fluoroquinolone into granulomas. Biochem Cell Biol. 2021 Feb;99(1):73-80.

 Kawasaki Y, et al. Inhibitory effects of bovine lactoferrin on the adherence of enterotoxigenic Escherichia coli to host cells. Biosci Biotechnol Biochem 64: 348-354, 2000.

de Araujo AN, Giugliano LG. Lactoferrin and free secretory component of human milk inhibit the adhesion of enteropathogenic Escherichia coli to HeLa cells. BMC Microbiol 1: 25, 2001

 Qiu J,et al.. Human milk lactoferrin inactivates two putative colonization factors expressed by Haemophilus influenzae. Proc Natl Acad Sci USA 95: 12641-12646, 1998.

Al-Mogbel MS, Godfred A Menezes , Mohamed T Elabbasy , Manal M Alkhulaifi , Ashfaque Hossain , Mushtaq A Khan Effect of Synergistic Action of Bovine Lactoferrin with Antibiotics on Drug Resistant Bacterial Pathogens Medicina (Kaunas). 2021 Apr 2;57(4):343.

 Zimecki M et al., The concerted action of lactoferrin and bacteriophages in the clearance of bacteria in sublethally infected mice.  Post. Hig. Med. Dosw. (Online). 2008 Feb 7;62:42-6.

minor remarks

a part of citations in the text is shifted by one reference, for example 69 should be 68.

lines 68-69: the explanation should be more understandable, e.g. “presence  of  CO32-  in iron binding site enables binding of Fe3, that makes lactoferrin pink in color”.

Figure 2:  wrong citation

line128: native lactoferrin in organism’s fluids is  iron- saturated in about 10-20%

lines 253 and 283: 6 and 15 aminoacids?

Figure 4:  the description not completed

Line 672: lactoferrin and not lactoferricin

The literature from 167 to 176 does not belong to main body text

Author Response

Thank you for revision of our review article and for your suggestions and remarks. On your initiative we supplemented article with missing information about antiparasitic activity of lactoferrin and lactoferrin derived peptides and indirect antibacterial activity of lactoferrin. Antibacterial activity of lactoferrin is described in separate paragraph (line 822), while indirect mechanisms of antibacterial activity are described under other mechanisms of lactoferrin antibacterial activity (line 224), where some of them were already mentioned. Minor remarks, that you have pointed out, were also fixed. Regarding your observation about literature that is not mentioned in the text, we would like to explain that this literature only refers to supplementary table 1. These articles mainly contain information about lactoferrin and/or lactoferrin derived peptide antimicrobial activity against certain microbe, but not containing any additional information about mechanism behind it, therefore we only use them in the supplementary table. To avoid ambiguities, we marked this in the reference list.

Reviewer 2 Report

The manuscript by Spela Gruden and Natasa Poklar Ulrih entitled “Diverse mechanisms of antimicrobial activities of lactoferrins, lactoferricins, and other lactoferrin-derived peptides” is an interesting review of lactoferrin biological properties. Over the last decade lactoferrin has been recognized as a major immune modulator relevant to the establishment and maintenance of immune homeostasis in humans. Many articles have recently been published on multifunctional properties of lactoferrin, in particular on antimicrobial activities/mechanisms. It is unclear why these articles are not discussed in the manuscript under review? The authors have attempted to review mostly historical data with fewer emphases on important recent reports.

Here is a short list of references that are missing and have to be further discussed in this review manuscript:

  1. Bruni et al. Antimicrobial Activity of Lactoferrin-Related Peptides and Applications in Human and Veterinary Medicine. Molecules. 2016 Jun 11;21(6):752
  2. Campione et al. Lactoferrin as Protective Natural Barrier of Respiratory and Intestinal Mucosa against Coronavirus Infection and Inflammation Int. J. Mol. Sci. 2020, 21,
  3. Lepanto et al., Lactoferrin in Aseptic and Septic Inflammation. Molecules 2019, 24, 1323.
  4. Kruzel et al., Lactoferrin in a Context of Inflammation-Induced Pathology. Front Immunol. 2017 Nov 6;8:1438.
  5. Sienkiewicz et al. Lactoferrin: an overview of its main functions, immunomodulatory and antimicrobial role, and clinical significance. Critical Reviews in Food Science and Nutrition, 2021. DOI: 10.1080/10408398.2021.1895063
  6. Zimecki et al., The potential for Lactoferrin to reduce SARS-CoV-2 induced cytokine storm. Int Immunopharmacol . 2021 Mar 12;95:107571.
  7. Actor et al.. Lactoferrin as a natural immune modulator. Curr Pharm Des. 2009;15(17):1956-73.

Having said that, the manuscript is of interest however more in-depth molecular mechanisms of lactoferrin antimicrobial activity need to be presented. Here are my concerns, questions, and suggestions:

GENERAL:

  1. Using consistently a plural form for lactoferrin is not justified, especially when the effects are different for individual molecules (e.g. human versus bovine lactoferrin).
  2. The literature is composing of rather historical reports on lactoferrin and clearly missing the important recent publications and similar works. Generally, there are too many references for the specific observations included.

SPECIFIC:

  1. (Line 10) The authors state that …“Among these, their antimicrobial activity has been the most studied, although the mechanism behind this remains to be elucidated”.…, so do we know any of these mechanisms? How about the chelation of iron by lactoferrin and subsequent protection against oxidative damage, chronic conditions, infections, trauma …?  Some of these mechanisms are well-established, thus do not need to be elucidated.
  2. (Line 18) Also, … “The role of their structure, amino-acid composition, conformation, charge, hydrophobicity, and others factors that affect their mechanisms of antimicrobial activity are also reviewed” …. Unfortunately, the structure/function is not well presented here and often referenced to the original reports with only brief conclusion. When comparing two compounds, e.g. lactoferrin with lactoferricin peptide, it is not enough to say that one has better antimicrobial activity than the other one; it should be stated by numbers how much better. In fact, this is consistently missing from the manuscript.
  3. (Line 24) … “Lactoferrins are also known as ‘red milk protein“ … it is very archaic expression, and not even properly explained here. Why red, or rather pink?; altogether not important.
  4. (Line 82) … Enzymatic digestion of bovine lactoferrin with pepsin by Tomita et al. in 1991 led to the discovery of cleavage peptides known as bovine lactoferricin (17-41) … check the references.
  5. (Line 101) … “The bacteriostatic activity of lactoferrins arises through their iron binding“ ….. Yes, and the role of lactoferrin in managing toxic iron is well beyond the bacteriostatic activity that needs to be better explained in this section. From the mechanistic point of view the antimicrobial activity of lactoferrin is either iron-dependent or iron-independent. All other mechanisms are clearly based on one:one interaction of lactoferrin with the specific structure of organism under discussion. Also, what is the function of lactoferrin sugars (glycans) in this context?
  6. Line 469) …Antiviral activities of lactoferrin …. A potential role of lactoferrin in recent pandemic of SARS-CoV-2 is missing.

Overall the manuscript is of important value and worth revision to make it compatible with the mission of IJMS.

Author Response

Thank you for revision of our review article and for your suggestions and remarks. On your initiative we supplemented article with missing information such as role of glycosylation in antibacterial activity of lactoferrin (line 72 and 228) and role and/or antiviral activity of lactoferrin in recent SARS COVID-19 pandemic (line 695). Furthermore, we made some corrections (line 10, 70 and 142) to avoid ambiguities. Here we would like to explain that sentence in line 10 only refers to antimicrobial activity of lactoferrin (e. g. we try to elucidate mechanism of antimicrobial activity) not others. Expression ‘’red milk protein’’ was one of the first names given to lactoferrin by researchers and we didn’t make it up (explained in line 70). 

As you indicated, recently many articles have been published on lactoferrin multifunctional properties, however most of these articles (especially the ones regarding antimicrobial lactoferrin functions) are review type. Furthermore, all of the articles that you proposed are review type and some of them don’t even refer to our review article. However, proposed articles were interesting and were therefore mentioned in our article (line 90).

However, some of your suggestions we can not consider and here is our explanation.

First of all, as you pointed out, many of our literature is old because we aimed to cite mostly original research articles. Since lactoferrin antimicrobial activity was one of the first discovered and studied, therefore this is no surprise. Furthermore, most of the resent studies are also focused on anti-cancer or immunomodulatory activity of lactoferrin and not so much on its antibacterial activity.

Secondly, one of your concerns was also that ‘’it should be stated by number how much better is one compound in compassion to another’’. Here we would like to explain that we intestinally avoided citing numbers, because stating that lactoferricin have better antimicrobial activity that native protein or that bovine lactoferricin is more potent than human lactoferricin in a general observation that was demonstrated by different articles where activity was different or they even used multiple or different microorganisms. We believe that citing numbers is not relevant an it will only confused reader.

At last, you mentioned that structure/function is not well represented. Here we would like to point out that our aim was to clarify some aspects behind the mechanisms of lactoferrin and lactoferricin antimicrobial activities. Since these mechanisms are also related with structural aspect (structure, amino-acid composition etc.), we consequentially included that in our review article. We represented structural difference between apo- an holo-form of lactoferrin and why holo-form lacks antimicrobial activity. We also included important observations from different studies that discussed lactoferricin structure, amino-acid composition etc. and its antibacterial activity.

There is a lot of literature available regarding natural antimicrobial peptides (AMP), however these peptides differ in their mechanisms of antimicrobial activity and also their structural aspect (α-helix and β-sheet peptides, cationic/anionic peptides, important residues etc.). Since there is a lot of antimicrobial proteins and peptides, our aim was to focus solely on antimicrobial activity of lactoferrin and lactoferricin derived peptides to elucidate its mechanisms of antimicrobial activity and structural aspect. We therefore mostly used only literature that refers to lactoferrin/lactoferrin derived peptides and didn’t discussed antimicrobial activity or structure of others antimicrobial protein/peptides. Also, some structure/function relationship in antimicrobial peptides are still unclear. We believe that article gives more than enough information regarding structure/function relationship, although main topic refers to mechanisms of antimicrobial activity.

Round 2

Reviewer 2 Report

Here are a few comments to the revised version of the manuscript by Spela Gruden and Natasa Poklar Ulrih entitled “Diverse mechanisms of antimicrobial activities of lactoferrins, lactoferricins, and other lactoferrin-derived peptides”. The most important critiques have been addressed and implemented. It would be advisable however to clarify the following:

  1. (Line 172) … “Furthermore, the release of LPS by lactoferrins can be blocked by high levels of divalent ions, such as Ca2+ and Mg2+ [57-59, 67]“ … release of LPS from where? … is missing, and similarly

  1. (Line 221) … “By releasing LPS, lactoferrins increase the permeability of the outer …” again, releasing from where? Perhaps a sentence or two should be added to explain the affinity of lactoferrin to LPS in bacterial wall of Gram negative bacteria.

  1. Is Fig 1 adopted from other papers? If yes this needs to recognized by proper references.

Overall, taking into consideration these minor corrections, the manuscript would be acceptable for final decision by the editorial office.

Author Response

As you proposed we clarify the Line 172 and 221. Regarding figure 1, we would like to inform you that this figure was made by our self in Schrödinger program Maestro 12.7.